# Measuring Child Labor: The Who's, the Where's, the When's, and the Why's

**Guilherme Lichand**[1]*, **Sharon Wolf**[2]

**1** Graduate School of Education, Stanford University, Stanford, California, United States of America,
**2** Graduate School of Education, University of Pennsylvania, Philadelphia, Pennsylvania, United States of America

☯ These authors contributed equally to this work.

* glichand@stanford.edu

## Abstract

Measuring child labor accurately is a major challenge: parents' and children's reports tend to differ dramatically, and there is typically no way to verify whose reports are truthful (if any). To overcome this challenge, this paper uses novel data from a cocoa certifier in Côte d'Ivoire that draws on satellite imagery to minimize under-reporting. Concretely, aerial photos allow them to select remote and hard-to-reach communities—where parents typically have not been sensitized by government or NGOs, averting social desirability biases—and to visit these communities while cocoa is being harvested—precisely when children in employment are very visible, making it easier for enumerators to impute it if parents still fail to report it. We compare their figures with those obtained from business-as-usual surveys with parents and children in these regions, and find that (1) reporting inconsistencies between parents and their children in fact decrease with household remoteness; (2) adults dramatically under-report child labor relative to the certifier data, by a factor of at least 60%; and (3) in turn, children self-reports are statistically identical to the latter. Taking advantage of an experiment that randomly assigned a text-message campaign to discourage child labor, we further show that parents' reports not only underestimate its prevalence, but can even lead to the wrong conclusions about the effects of policy interventions.

## 1 Introduction

Child labor is a pervasive practice in agriculture, especially in West Africa, where the global cocoa industry sources roughly half of its produce. Defined according to international standards and national legislation, child labor typically characterizes any work by children under the age of 12, and excessive or risky work by 12–17 year-olds (see Section 2.1.1). 2020 data from the International Labor Organization (ILO) documented 160 million child workers worldwide—9.6% of all 5–17 year-old children [1]. Strikingly, that figure was over 2-fold in Sub-Saharan Africa (23.9%). In Côte d'Ivoire, the setting of our study, 15% of the cocoa industry employees that year were children [2].

While such figures have mobilized massive global attention and resources to combat child labor, a key challenge in interpreting them is reliability. This is because while official statistics

**Data availability statement:** A complete replication package is available at https://osf.io/uab5c.

**Funding:** Study conducted in partnership with Transforming Education in Cocoa Communities (TRECC) and generously funded by the Jacobs Foundation Science of Learning initiative. The funders had no role in study design, data collection and analysis, decision to publish, or preparation of the manuscript. Any views and opinions contained in this paper are those of the authors and do not necessarily reflect the views or opinions of TRECC or the Jacobs Foundation.

**Competing interests:** Lichand is a partner and chairman at Movva, the implementing partner of the intervention evaluated as part of this study. This does not alter our adherence to PLOS ONE policies on sharing data and materials. Wolf has no competing interests to declare.

on child labor are essentially based on adult reports (see Section 2.1.1), there are many reasons why parents or caregivers may fail to truthfully disclose children in employment. They might fear that admitting to child labor (e.g., that their under-12 year-old children help during the harvest season) could lead them to face legal action from child protective services, or adversely impact their livelihoods (e.g., if the companies sourcing from them are punished for child labor in their supply chain, which could trickle down to lower prices or even to discontinuation of their income; see [3]). Importantly, if parents omit children in employment, child labor will be under-estimated in the data—since child labor is *contained* in children in employment.

This paper uses novel data from a cocoa certifier in Côte d'Ivoire to overcome this challenge. ENVERITAS, a global NGO that certifies smallholder coffee and cocoa farms when it comes to agricultural best practices—including the absence of child labor –, draws on satellite imagery to minimize under-reporting. Concretely, aerial photos allow them (1) to select remote and hard-to-reach communities, where parents typically have not been sensitized by government or NGO campaigns to discourage child labor, averting social desirability biases; and (2) to visit these communities during harvest season, precisely when child labor is very visible, making it easier for enumerators to mark it as present even if parents still fail to report it. We compare their figures with those obtained from business-as-usual surveys with parents and children in these regions, which partially overlap with the remote villages where ENVERITAS certifies farming practices. Our sample focuses on primary school children, nearly all under 12 years old—for whom any hours in employment are classified as child labor by national legislation.

Before turning to the certifier data, we start by documenting patterns in reporting discrepancies in our own surveys that lend credibility to our empirical strategy. Restricting attention to households in our sample with valid GPS location data, 29.3% of adults admit that their children worked at least an hour in cocoa fields in the month before the survey. In contrast, 51.8% of children report that they did so—a striking 77% average difference in reporting rates. First, we document that such discrepancies indeed decrease with the distance from each household to the school their children attend: the difference between children's and parents' reports decrease significantly with every additional km from the school, consistent with ENVERITAS's claim that village remoteness is inversely related to reporting biases. Second, we estimate that children's self-reports do *not* increase with distance from the school, minimizing concerns with child labor in more remote villages misrepresenting its typical prevalence. We further document that not only are discrepancies between children's and parents' reports much lower when it comes to domestic work—a more socially acceptable activity and typically not the subject of advocacy campaigns, in contrast to labor in cocoa fields –, but also, that such discrepancies do *not* decrease with household remoteness. All in all, results back up the certifiers' sampling strategy as a way of measuring the ground truth for the prevalence of child labor, which we can then use to validate parents' and children's reports in the sub-sample for which the two datasets overlap.

Our main contribution is to document first-hand that adult surveys indeed under-report the prevalence of child labor, and the extent of under-reporting. In regions with subsequent third-party verification, 45.5% of children reported having worked in cocoa plantations in the previous month, matching almost exactly the 47.8% prevalence indicated by the certifier; in contrast, only 16.2% of parents in those regions reported children in employment—a nearly 2/3 reporting gap. Across regions, under-reporting ranged from 60% to 85%.

Previous studies were not able to document the extent of under-reporting by parents. These papers rely on three types of comparisons. First, comparisons between parents' reports under different conditions for social desirability bias [e.g.,4], to document that such biases

exist—but unable to pin down their magnitude. Second, comparisons between adults' and children's reports [5,6], to document that discrepancies exist—but unable to pin down what the 'ground truth' is. Third, comparisons between an objective measure of children in employment and children's self-reports [7], to document that the latter are accurate—but unable to pin down under-reporting by parents. Similarly to ours, the latter study documents that children accurately report the number of hours they work, using logs from GPS trackers worn by different household members to verify data from surveys and activity diaries. The study, however, actively refrained from having enumerators ask adults about child labor. In the absence of parents' reports, it could not document whether parents under-report children in employment and, if so, by how much.

Our second contribution is to document that basing child labor accounts on surveys with parents not only underestimates its prevalence, but can also bias evaluations of how it responds to policy interventions. Taking advantage of a randomized control trial that assigned some Ivorian parents to messages discouraging child labor in cocoa fields to study whether the estimated impact of the intervention on child labor depends on how the latter is measured, we find that while messages had no effect on children in employment according to children themselves, they significantly increased children in employment according to parents (by 55.1%). Presumably, the reason for such discrepancy is that the intervention tried to foster investments in children that would reduce children's participation in labor activities *without explicitly condemning child labor*—partially deterring social desirability biases. Once again consistent with the claim that such biases are stronger in communities that have been previously sensitized, treatment effects on child labor according to parents sharply decay with our measure of remoteness.

These findings have key implications for how child labor should be measured. One possibility is to survey children directly (the '*who*'). Our results indicate that this would most likely yield accurate estimates of children in employment, but it might also involve complex technical and ethical dilemmas. It might be hard to ensure a unified understanding of what characterizes employment, especially among younger respondents. Most importantly, participation might put children at risk if it triggers backlash by parents. An alternative is to survey adults, as usual, but to focus on hard-to-reach communities (the '*where*') during harvest season (the '*when*')—leveraging technological advances such as satellite imagery. We further discuss the implications and limitations of our findings in Section 4.

## 2 Materials and methods

### 2.1 How child labor is measured

There are striking differences between children's and adults' reports in the few settings where both have been asked independently about children in employment. According to NORC (a research institute based at the University of Chicago), which surveys children directly about the number of hours worked in cocoa fields, 38% of 5-17 year olds in Côte d'Ivoire reported to have worked in 2018-19 [2]; in contrast, the ILO figure for 2016—based on adult reports for children employment—was only 23% [1]. Even worse, different sources tell very different stories about child labor's recent trends. While ILO data indicate a 38% decrease in child labor worldwide since 2000 [8], NORC data record a nearly 65% *increase* in child labor since 2008-09 [2].

While these differences are suggestive that official statistics based on adult reports are biased, it is hard to be sure; besides differences in whom is surveyed, discrepancies across surveys might also accrue to differences in their geographical coverage or in the timing of data

collection, or to other methodological differences. Moreover, in the absence of verification, it is unclear whether children's self-reports do not suffer from reporting biases too.

**2.1.1 ILO methodology.** The International Labour Organization (ILO) follows the Convention on the Rights of the Child, the ILO Minimum Age for Admission to Employment Convention (No. 138), and the ILO Worst Forms of Child Labour Convention (No. 182) to define child labor [1]. According to these conventions, whether children in employment characterizes child labor depends on the child's age, the number of hours dedicated to work, and the work conditions. For children 11 years old or younger, any employment characterizes child labor. For those between 12 and 14 years old, 15 or more weekly work hours and/or hazardous work conditions (for any number of hours) characterize child labor. Last, for those between 15 and 17 years old, child labor applies in case of 43 or more weekly work hours and/or hazardous work conditions. In SI Text, we illustrate all conditions used by the ILO to define child labor for children of different age groups. Importantly, the ILO encourages countries to modify and complement these guidelines with national legislation that makes figures contextually appropriate. In Brazil, for example, local laws establish that any employment characterizes child labor for children 14 years old or younger—a stricter standard relative to the ILO guidelines.

Statistics on children in employment, number of hours worked, and work conditions come from different surveys around the globe. The ILO does not collect the data itself but, rather, harmonizes data from these different sources to compute the prevalence of child labor according to its methodology. All leading international organizations currently follow the ILO methodology. For example, UNICEF adapted its Multiple Indicator Cluster surveys after 2013 to match ILO guidelines. The World Bank also tracks child labor following the same methodology, only for 7-14 year-olds rather than 5-17 year-olds.

In the ILO methodology, child labor is computed by *combining* surveys with adults and those with children. Concretely, the ILO methodology *only uses the adult questionnaire* to compute children in employment. In turn, the children questionnaire is used to assess hazardous work conditions *only among those who are employed* ("[a]s in the previous rounds, the current round of the Global Estimates of Child Labor uses data obtained from the adult questionnaire, except for conditions of work, where the information from the child questionnaire is deemed to be more reliable"; [9], p. 59). In other words, child labor is a subset of children in employment—which is exclusively based on parents' reports. As such, official statistics on child labor depend crucially on the accuracy of the latter.

**2.1.2 NORC methodology.** NORC, a research institution at the University of Chicago, has tracked child labor in the cocoa industry for Ghana and Côte d'Ivoire since 2015, building on Tulane University's work in the region dating back to 2008. It reports statistics associated with child labor, for cocoa production in particular and for agricultural activities more broadly [2]. NORC defines child labor based on the number of work hours and work conditions for children of different age groups, consistent with the ILO methodology (see SI Text). Different from the ILO, however, NORC surveys children about the number of hours worked *directly*, and defines child labor based on *children's self-reports* ("[u]sing the responses of children relating to engagement in cocoa production, we generated estimates of children's engagement in child labor and in hazardous child labor in cocoa production-related activities"; [2], p.61).

Comparing the different data sources—which differ according to the reporting sources used for computing children in employment—is telling. 2016 ILO data for Côte d'Ivoire (https://ilostat.ilo.org/topics/child-labour/) indicated that 17.5% of 5-17 year-old children engaged in economic activity and household chores. In contrast, NORC data indicated that,

as recently as 2018-19, 64% of all 5-17 year-olds in cocoa-producing regions of Côte d'Ivoire worked in the past 7 days, and 78% in the past 12 months.

While differences are striking, the NORC surveys cover different geographies and years than those used to compute official statistics, making it hard to attribute the gaps to under-reporting by adults in the ILO, UNICEF and World Bank data. Moreover, even if one would accept that parents report children in employment to a lesser extent than children themselves (see, for instance, [5,6]) or that parents' reports are prone to social desirability biases [4], it could be that children's self-reports are similarly unreliable. Without additional data to *verify the actual prevalence* of children in employment, one simply cannot tell.

**2.1.3 ENVERITAS methodology.** ENVERITAS is a not-for-profit NGO that certifies coffee and (more recently) cocoa companies by verifying farming practices in their supply chain—from chemical usage to child labor (see, for instance, [10]). Since 2020, their methodology has been aligned with the ILO standards.

The certifier has specialized in reaching small, hard-to-reach and remote farmers, who often cannot even be located by traditional certifiers. To do that, ENVERITAS relies on partnerships to access fresh acquisitions of 50cm-resolution satellite data filtered for quality (with alleged maximum cloud covers of 15%, "crucial for finding (...) farms in cloudy equatorial regions"; [11]), combined with machine learning models to identify specific crops. Additional details on how such models have been successfully used to map coffee-growing households in different geographies can be found in [12]. In Côte d'Ivoire, the certifier has applied this methodology to identify constraints to quality education and early childhood development—including child labor—in cocoa-growing communities [13].

The combination of satellite imagery with machine learning models generates GPS coordinates for each ENVERITAS field team. Pins in Google Maps assign plots to be surveyed to each enumerator. Michael Kra, country lead for ENVERITAS, points out that "*When one surveys farmers, sometimes they do not tell you the truth. The pins will tell us to go where the truth is*". The idea is that farmers in hard-to-reach communities are less sensitive to social desirability biases. Michael Kra explains: "*We work in very rural communities. The road is often not good, requiring enumerators to use boats to cross rivers, motorcycles, etc. These communities being much less accessible, farmers have been rarely sensitized about child labor. This is why it is easier for them to openly disclose child labor. We can only find these communities because of satellite imagery*". SI Text includes a sample satellite picture to illustrate how they are able to locate cocoa plantations based on aerial imagery. SI Text also compiles pictures from enumerators in the field as they survey some of these remote communities, documenting how they often have to cross dirt roads and flooded paths in order to reach them.

ENVERITAS further monitors cocoa harvests in the country with the help of satellite imagery, surveying farmers specifically during harvest season. With harvesting activities ongoing at the time of the survey, the potential presence of child labor becomes apparent: not only it is much harder for parents to falsely deny children in employment when it is more visible, but also, ENVERITAS complements survey data with direct field observation [12]. Michael Kra explains how surveying farmers during harvest time make it harder to under-report child labor: "*Enumerators can mark child labor as present even if a parent fails to report it (although this happens in less than 25% of surveys). They can add comments with additional information they have observed in the farm: e.g., 'child is helping, climbing cocoa trees'*".

## 2.2 Background for this study

**2.2.1 Study sample.** Our study takes place in the cocoa-producing regions of Aboisso and Bouafle in Côte d'Ivoire. Along with Ghana, the country hosts almost 2/3 of the world's cocoa

production. This has been linked to one of the highest incidences of child labor worldwide, with nearly 1.6 million children employed in cocoa fields (https://foodtank.com/news/2021/02/norc-report).

We collected child labor data in the context of a broader research project, focused on evaluating different communication interventions to prevent student dropouts (see [14]). As such, we focus on a sample of primary school children, all of whom were enrolled at baseline. We discuss the implications of that sampling restriction for the generalizability of our findings in Section 4.

SI Text provides descriptive statistics for our study sample. Almost all (92%) of participating children are 5-11 years old—for whom any form of employment is considered child labor according to ILO guidelines. Half of children in our sample are girls and live in rural areas (defined according to their parents' income source), and slightly over half of them are enrolled in the first primary cycle (CP2). Nearly a quarter (22%) of households in our sample are extremely poor—at baseline, they made at most a little over 1 USD a day. For only 18% of households income from all sources was more than 6 USD a day at baseline.

**2.2.2 Campaign to discourage child labor.** [14] evaluated a communication campaign (Eduq+, implemented by Brazilian EdTech Movva) that delivered nudges (reminders and motivational messages meant to make children's school life 'top of mind') directly to parents' mobile phones. The intervention was implemented over the 2018-19 school year. Nudges were organized in thematic sequences – comprised of four messages –, with two messages delivered each week. Content was catered to each students' age group. Messages tried to encourage parents to participate more actively in their children's school life. There was some emphasis on showing up to school, especially in the context of parent-teacher meetings, and on discouraging harmful practices like corporal punishment as a disciplining strategy.

Several sequences explicitly discouraged child labor in cocoa fields, describing how it might detract from child development and learning. The language was careful, in an attempt to openly discuss the issues without creating stigma or setting social expectations that ultimately make it harder to track whether children work on the fields. To illustrate the approach, during the intervention, parents received a text message stating "*It is important that you child complete her/his education! School can provide a better future not only for her/him, but for your whole family*". A few days later, another text encouraged them to "*Talk to your child about the importance of focusing on her/his education. Equally important as learning family traditions is learning the values and skills that only the school can teach*". See [15] and [14] for additional details about the campaign.

In the experiment, nudges to parents were cross-randomized with nudges to teachers, aimed at increasing their attendance and time-on-task while teaching. For simplicity, we focus on discrepancies between parents' and children's reports between the treatment condition that had only parents nudged and the control group (which did not receive any messages) in the main text. SI Text compiles results for the other treatment arms.

## 2.3 Data and outcomes

**2.3.1 Survey data.** This research was approved by the University of Zürich Institutional Review Board (Protocols OEC IRB # 2018-035 and OEC IRB # 2019-052). Côte d'Ivoire did not require local IRB approval for non-medical research at the time. Our study comprises 198 CP2 and CE2 classrooms (second and fourth grades, respectively) across 99 Ivorian public schools in the cocoa-producing regions of Aboisso and Bouafle. Within each school, we randomly drew 13 CP2 students and 12 CE2 students to be surveyed at baseline (at the beginning of the school year, from 01/10/2018) and end line (at the end of the school year, from

01/06/2019). Importantly, children and parents were surveyed independently. Enumerators ensured that this was the case, especially since we also tested children's numeracy and literacy skills as part of the broader project, which required them to sit by themselves—only accompanied by our survey team.

We also conducted an additional follow-up survey at the beginning of the following school year (from 01/10/2019). In this follow-up, we surveyed all teachers and only one parent per classroom, but no children. This follow-up data focused on collecting additional information about work conditions for children in employment, but it also provides a measure of children in employment according to parents already after the growing season [16] and much closer to the timeline of the ENVERITAS data collection. Data were accessed for research purposes in 01/12/2019.

The sample comprises 1,285 CP2 students and 1,190 CE2 students surveyed at baseline along with their parents, in addition to 198 teachers. We were able to track all teachers, 1,157 CP2 students (90.0%) and 1,086 CE2 students (91.3%) at end line. We assigned replacement households in case the ones drawn could not be tracked by enumerators. All participants or their legal guardian(s) verbally consented to participate; minors still had to assent. All participants were free to discontinue their participation at any point without sanctions. At the end line, no children—and less than 3.5% of their parents—refused to answer about their employment conditions. We discard child-parent pairs involving refusals, focusing on the 2,500 observations for whom we can compare children's and parents' reports. Out of those, we have information on all baseline characteristics that we use as controls in some of our specifications for 2,246 observations. Furthermore, we have information on home GPS location (which we use to compute a measure of remoteness) for 1,790 of those. Missing data for the latter often involves parents and children whom were surveyed at the school or in some other location.

SI Text compiles the survey questions related to children in employment in each wave. As indicated, we asked parents and children the same question about children in employment in cocoa fields at both baseline and end line ("*In the last month, have you [any of your children] engaged in one or more of the following activities, for one hour or more? Work in a cocoa plantation*"). This aligned with the data collected by ENVERITAS and NORC, and follows the ILO methodology for measuring children in employment. Since parents are not asked *specifically* about the child from whom we elicited employment information independently, that should decrease our ability to detect under-reporting by parents (e.g., if some answer affirmatively about older children, or about children who are already out of school).

Even though we also surveyed parents and children about other forms of employment (e.g., domestic work and construction), we do not analyze these data in this paper because we have no way to verify such reports. These additional measures of children in employment are described and analyzed in [14].

**2.3.2 Certifier data.** We use data on children in employment collected by ENVERITAS over January 2020, during the harvest season in Aboisso and in two sub-regions of Bouafle (Bouafle 2 and Tiapoum Adiake). ENVERITAS's sampling frame relied on geographical units of 10,000 farmers, identified via satellite imagery. They randomly drew 125 farmers to be surveyed in each unit. 8,150 households were approached by ENVERITAS, 7,402 of which were successfully surveyed. The certifier also surveyed schools in regions outside of our study sample; see SI Text. We do not use data in these other regions, or collected prior to 2020 (before the survey instrument was consistent with the ILO methodology regarding the definition of child labor).

Adults surveyed by ENVERITAS were asked "*Do any of your children between 6 and 16 years old help you work on the cocoa farm?*". Children were not surveyed directly by ENVERITAS; thus far, they have only piloted surveys with children in Tonkpi, a region outside our study sample.

## 2.4 Empirical strategy

### 2.4.1 Assessing the certifier claim that remoteness is inversely related to reporting biases.

To assess whether remoteness is indeed inversely related to reporting biases, we estimate how children's and parents' reports vary with their household's distance to the school (in km). SI Text includes the histogram for this measure, computed as the linear distance between the school and home GPS locations. 70% of households with valid home GPS data are within 1km of the school. While some are as far as 20km from the latter, the vast majority are within a 2.5 km radius of it.

Taking advantage of natural variation in distance to the school, we estimate a linear model for the association between distance and children's reports, and between the former and the discrepancy between children's and parents' reports. We restrict attention to endline data, since that survey was the closest to harvest time—precisely when child labor is supposed to take place and, hence, around the same time the certifier enumerators were also in the field. As such, we further drop from the analyses households targeted by nudges to parents, as we later document treatment effects on reporting discrepancies.

When analyzing discrepancies, we also control for the school-level share of children in employment, and allow its coefficient to vary with whether the child has reported to work—all of which might influence the extent of reporting biases. We assess the sensitivity of our estimates to outlier observations and to specification choices by plotting the relation between reporting discrepancies and distance from the school, dropping observations further than 1.5 km from the school.

Importantly, we estimate those relations not only for child labor in cocoa fields, but also, for domestic work (helping with household chores, etc.) —much less sensitive to social stigma, and which has not been systematically targeted by NGOs and international organizations promoting children and adolescents' rights in the region. Domestic work provides a counterfactual to assess the claim that discrepancies between parents' and children's reports are indeed driven by social desirability biases, and a placebo test for the claim that village remoteness should decrease the extent of reporting biases by the adults. Concretely, we hypothesize that (1) differences in children in employment across parents' and children's reports should be significantly lower for domestic work relative to labor in cocoa fields, and that (2) the magnitude of discrepancies should decrease with the distance to the school only for labor in cocoa fields.

### 2.4.2 Assessing the accuracy of different reporting sources.

We assess the accuracy of children in employment reports according to parents and according to children themselves by comparing our survey data to the certifier data within the regions for which the two datasets overlap.

We report p-values from tests of differences in proportions of children in employment according to each source, considering equal population variances when comparing parents' and children's reports (through an Ordinary Least Squares regression, given the paired design), and unequal population variances when comparing any of them to certifier data—since the latter was collected from a different sample. We cluster standard errors at the regional level.

### 2.4.3 Assessing the sensitivity of the campaign's effect sizes to different reporting sources.

To document whether different reporting sources might lead to bias in evaluating the impacts of interventions to discourage child labor, we contrast effect sizes of nudges to parents on the prevalence of children in employment in cocoa fields based on children's self-reports or on adult reports. We also allow treatment effects based on parents' reports to vary with the distance to the school, to assess whether results are consistent with the spatial patterns we document for discrepancies between parents' and children's reports.

## 3 Results and discussion

### 3.1 Descriptive statistics based on children's self-reports

SI Text documents the aggregate prevalence of children in employment, and that by student characteristics, according to children's self-reports at baseline. 38.1% of children reported to have worked at least one hour in cocoa fields over the previous month. As a point of comparison, by the end of the school year (closer to harvest season), this figure was up to 50% (see SI Text). The prevalence of child labor was only slightly higher among fourth graders than that among second graders (39.6% vs. 36.8% at baseline, and 51% vs. 48% at end line). Boys were nearly 50% more likely to work in cocoa fields at baseline than girls (44.4% vs. 31.2%). The baseline prevalence of children in employment was higher for the bottom income bracket (38.9%), but not low even for the top income bracket (23.9%). Naturally, child employment in cocoa fields was much higher in rural areas (52.4% vs. 23.9%). For an account of adult reports about children's work conditions, elicited in the follow-up survey, see SI Text.

SI Text also illustrates correlations between classroom-level prevalence of child labor and educational outcomes. Consistent with common sense, classrooms with a higher share of children in employment at baseline feature lower test scores by the beginning of the school year, and higher dropout rates over the course of the school year. Estimating a linear relation between the variables in each case suggests that moving from 0% to 20% children in employment is associated with about 0.08 s.d. lower test scores—what children tend to learn in one school quarter, and the magnitude of effect sizes of many educational interventions, such as nudges to parents evaluated in this setting [15]. Similarly, moving from 0% to 40% children in employment is associated with roughly doubling dropout rates.

While these associations are not causal, they help understand the centrality of the issue for governments and international organizations monitoring children's rights. This is why accurate measurement is key.

### 3.2 Household remoteness and reporting discrepancies

We start by investigating the association between a measure of household remoteness—the distance from each household to the school where the child participating in the study was enrolled—with discrepancies between parents' and children's reports. For these analyses, we restrict attention to the 1,790 households in our sample with valid GPS location data, and drop observations targeted by nudges to parents (in combination with or independently of nudges to teachers)—as we later show that the intervention affects reporting (see Section 3.4) –, resulting in 1,363 observations.

If remote communities are indeed less often or less intensely sensitized, a challenge is that remoteness might affect both under-reporting *and* the true prevalence of child labor. To separate these potential effects, Table 1 starts by assessing whether children's self-reports vary systematically with our measure of remoteness (in column 1); and columns (2) to (4) assess how the discrepancy between children's and parents' reports varies with that measure. Panel

**Table 1. Association between household distance to school, children's self-reported employment, and within-household reporting discrepancies between parents and children.**

| | Children in employment (self-report) | Discrepancy (child - parent) | | |
|---|---|---|---|---|
| | (1) | (2) | (3) | (4) |
| **Panel A - Work in cocoa fields** | | | | |
| Distance from school (km) | 0.014 | -0.014 | -0.017** | -0.017*** |
| | (0.014) | (0.012) | (0.007) | (0.006) |
| Child labor (self-report) | | 0.633*** | 1.063*** | 1.073*** |
| | | (0.037) | (0.066) | (0.072) |
| Average child labor at school | | | -0.406*** | -0.388*** |
| | | | (0.078) | (0.086) |
| Child labor × | | | -0.512*** | -0.511*** |
| Average child labor | | | (0.122) | (0.131) |
| Sample mean | 0.506 | 0.229 | 0.229 | 0.228 |
| R-squared | 0.001 | 0.376 | 0.471 | 0.477 |
| Observations | 1363 | 1362 | 1362 | 1235 |
| **Panel B - Domestic work** | | | | |
| Distance from school (km) | -0.003 | 0.010 | 0.009 | 0.000 |
| | (0.007) | (0.012) | (0.011) | (0.009) |
| Domestic work (self-report) | | 0.617*** | 0.523 | 0.383 |
| | | (0.046) | (0.432) | (0.461) |
| Average domestic work at school | | | -0.412 | -0.485 |
| | | | (0.474) | (0.524) |
| Domestic work × | | | 0.136 | 0.339 |
| Average domestic work | | | (0.507) | (0.536) |
| Sample mean | 0.886 | 0.082 | 0.082 | 0.088 |
| R-squared | 0.000 | 0.212 | 0.215 | 0.243 |
| Observations | 1363 | 1361 | 1361 | 1234 |
| Baseline controls | No | No | No | Yes |

*Notes*: Endline data excluding households targeted by nudges to parents (in combination with or independently of nudges to teachers). In column (1), Children in employment (self-report) = 1 if the child answered affirmatively to the question: "In the last month, have you engaged in one or more of the following activities, for one hour or more? Work in a cocoa plantation" in Panel A, and "Domestic work, such as buying food or cooking, cleaning the house, do the laundry, take care of children or other sick or old relatives", in Panel B. In columns (2) to (4), Discrepancy (child - parent) = self-reported employment - parent's reported children in employment, whereby the latter = 1 if the parent answered affirmatively to the question "In the last month, has any of your children engaged in one or more of the following activities, for one hour or more? Work in a cocoa plantation", in Panel A, and "Domestic work, such as buying food or cooking, cleaning the house, do the laundry, take care of children or other sick or old relatives", in Panel B. Distance from school (km) is the linear distance between the household and their child's school GPS location, in kilometers. Average child labor at school and Average domestic work at school are the percentage of children surveyed in each school who answered affirmatively to each question. Baseline controls include child gender, grade indicators, standardized test scores (averaged across numeracy and literacy); and summary measures of parental engagement, student effort, socio-emotional skills, working memory, visual attention, impulsivity, self-esteem, and mindset (see [14]). Summary measures computed following [17], standardizing each component by normalizing values by the mean and standard deviation of the control group at the baseline survey within each grade, and then averaging over all standardized components. Sample restricted to observations with valid home GPS coordinates. All regressions estimated through Ordinary Least Squares, with standard errors clustered at the school level.
*** p<0.01, ** p<0.05, * p<0.10

A focuses on labor in cocoa fields; and Panel B turns to domestic labor. All columns control for children's self-reports; column (3) further controls for the school-level prevalence of child labor (according to children) and its interaction with each child's self-report, as those might also affect the extent of reporting biases; column (4) additionally controls for baseline characteristics. Because we include a school-level measure of prevalence, we cluster standard errors at the school level.

In Panel A, column (1) documents that distance from school is not associated with children's self-reports. The average end-line prevalence of children in employment within the sub-sample with valid home GPS data was 50.6%; such prevalence did not systematically increase with the household's distance to the school. This is consistent with the idea that more remote communities do not necessarily feature more children working in cocoa fields. In turn, columns (2) to (4) document that such distance was systematically associated with lower discrepancies between children's and parents' reports. While, in this sub-sample, the average discrepancy between children's and parent's reports was 22.9 p.p., we estimate that it decreased by 1.7 p.p. (significant at the 5% level) with every km away from school, 7.4% of the average discrepancy. Incidentally, other estimated coefficients are also informative: consistent with our previous discussion, the more common children in employment is at the school, the lower the reporting discrepancies—consistent with the role of social expectations. Results support the certifier claim that remoteness is inversely associated with reporting biases, without necessarily being associated with children in employment itself.

Fig 1 documents that the relation between household remoteness and reporting discrepancies is not an artifact of outlier observations or specification choices. Dropping observations further than 1.5 km from the school from the analyses, the binscatter plot shows that discrepancies indeed decline systematically with distance. If anything, the relation is even stronger within that range: a linear regression estimates that discrepancies decrease 4.5 p.p. per km, from a baseline of 25% (an effect size significant at the 10% level).

One concern is that our measure of remoteness captures some other underlying differences in these communities other than those linked to prior exposure to sensitization campaigns—and corresponding differences in the prevalence of social desirability biases as a result. To tackle that concern, Panel B of Table 1 replicates the previous analyses to domestic labor in

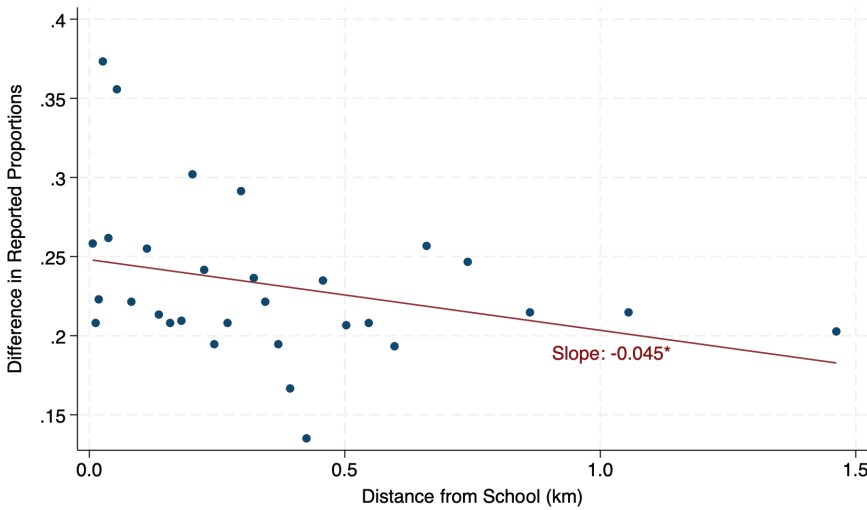

**Fig 1. Within-household discrepancies of children in employment in cocoa fields (self- vs. adult reports), by distance from school (in km).** *Notes:* The y-axis displays the difference between children in employment in cocoa fields reported by children themselves and that reported by their parents, in response to the question "In the last month, have you / has any of your children engaged in one or more of the following activities, for one hour or more? Work in a cocoa plantation". The x-axis displays distance from the child's home to the school (in kilometers). Observations with distance greater than or equal to 1.5 km were dropped, resulting in the exclusion of 12% of observations (490 households). Data divided into 30 quantiles. Fitted line estimated through Ordinary Least Squares (OLS), with no control variables; *** p<0.01, ** p<0.05, * p<0.10.

an attempt to rule that out. Importantly, S10 Fig in SI Text documents that, in the full sample, discrepancies are significantly lower when it comes to domestic labor. Here we are interested in whether such discrepancies are *not* negatively associated with our measure of household remoteness. Panel B documents that, different from labor in cocoa fields, differences between parents' and children's reports do not systematically change with the distance from the school.

All in all, results lend confidence to the claim that the certifier's sampling strategy might indeed reveal the 'ground truth' when it comes to the local prevalence of child labor. Moreover, they suggest children's self-reports in the remote villages surveyed by ENVERITAS are likely representative of those in other villages. As such, we can confidently use that data to validate the prevalence of child labor according to different reporting sources in the sample for which the two datasets overlap, in the next section.

## 3.3 Validating survey data with certifier data

We now turn to the comparisons between independent reports by parents and children in our end-line survey data, and ENVERITAS data. Fig 2 presents the prevalence of children in employment according to each reporting source, along with p-values for pair-wise statistical tests of differences in proportions. We restrict attention to the aggregate prevalence based on each reporting source across the regions where both datasets overlap in the main text. Moreover, we also restrict attention to households in the ENVERITAS data whose children were *all* enrolled in school at the time of the survey, to make its sampling frame consistent to that of our surveys. SI Text further documents comparisons within each region, and relaxing that sample restriction. Our findings are very robust across the different sets of comparisons.

In the absence of under-reporting, the rate of children in employment according to either parents' or children' reports should be identical (in fact, since the question directed to parents was about *any* of their children, the former should be weakly greater than the latter). This is, however, strictly at odds with what we find. Fig 2 documents that, in regions with subsequent verification, 45.5% of children reported to have worked in cocoa plantations in the previous month, matching almost exactly the 47.8% prevalence indicated by the certifier (p-value of the difference = 0.746). In contrast, only 16.2% of parents in those regions reported employing children—a nearly 2/3 reporting gap ($p$ = 0.000). S13 Fig in SI Text shows that, across regions, under-reporting by adults was striking, ranging from 50% to 90% ($p$ = 0.000 in each case).

**3.3.1 Robustness to differences in the timing of the surveys.** One potential caveat of the results above is that our end-line survey (when both children and parents were asked about children in employment) dates from June 2019, almost 6 months prior to the ENVERITAS data collection. Here we assess whether such difference matters for our findings.

To do that, we take advantage of our follow-up wave, conducted in October 2019 (much closer to January 2020, when ENVERITAS conducted its survey). While children were not surveyed in that wave, we can compare parents' reports across the end-line and follow-up surveys to gauge whether adult reports converged to children's self-reports as harvest season was approaching. SI Text documents that this was *not* the case. At the follow-up wave, only 28% of parents admitted that children worked at all in cocoa fields during the school year. This figure was still only about half that reported by children at the end-line survey—even though, at the follow-up wave, we asked parents to report on work performed by children in cocoa fields *at any point during the previous school year*. SI Text further documents that teacher reports of child labor over the course of the previous school year were statistically identical to parents' reports at the end line.

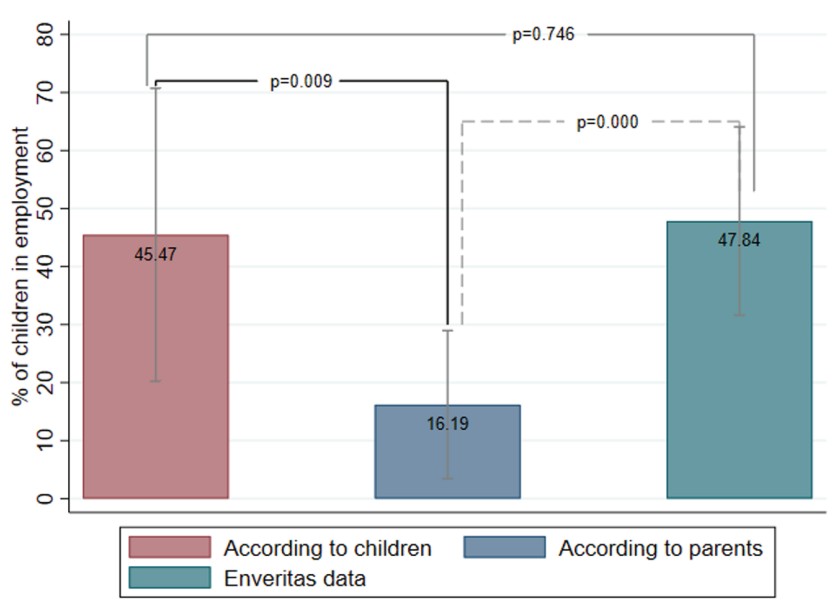

**Fig 2. Validation of child labor measures using third-party data, households with all children in school. Notes**:
Columns show the share of children who worked at least an hour in cocoa fields over the previous month, according
to children (in red), parents (in blue) and ENVERITAS (in green), along with 90% confidence bars. In the ENVERI-
TAS data, we restrict attention to households whose *all* children between 6 to 11 years old were enrolled in school at
the time of the survey (in SI Text, we relax that sample restriction). The figure reports the average prevalence across
all regions for which survey data overlaps with ENVERITAS data. Children answered the following question at end
line: "In the last month, have you engaged in one or more of the following activities, for one hour or more? Work in a
cocoa plantation", as described in S1 Table. Parents answered the following question at end line: "I will now ask you
some questions about activities that your children might have recently performed. In the last month, has any of your
children engaged in one or more of the following activities, for one hour or more? Work in a cocoa plantation", as
described in S1 Table. In the survey conducted by ENVERITAS during the harvest season (identified through satellite
imagery), farmers answered the following question: "Do any of your children between 6 and 16 years old help you
work on the cocoa farm?". For the measures reported by children and parents, observations are restricted to the con-
trol group of the communication intervention. P-values from tests of proportions with unequal population variances
(when children's or parents' reports are compared to ENVERITAS data; accounting for each source's intra-cluster
correlation computed at the regional level), and from tests of proportions with equal population variances (when
comparing children's and parents' reports; through Ordinary Least Squares regressions, clustering standard errors at
regional level). Sample sizes are as follows: (i) Children: 475; (ii) Parents: 475; and (iii) ENVERITAS: 232.

## 3.4 How adult reports can also bias policy evaluations

Last, we present evidence that relying on parents' reports for children in employment might
not only lead to inaccurate estimates about the prevalence of child labor, but also, to poten-
tially incorrect conclusions about the effects of interventions to discourage it. Table 2 esti-
mates treatment effects of nudges to parents on children in employment at end line, relative
to the control group. Column (1) uses children's self-reports as the dependent variable, while
columns (2) to (4) use parents' reports. Columns (1) and (2) use the full sample for whom
we have children's and parents' reports; columns (3) and (4) restrict attention to those with
valid home GPS information. Column (4) allows treatment effects to vary with our measure
of remoteness—the linear distance from the household to the school where the participating
child was enrolled (in km). The idea is that part of the effects of the intervention on parents'
reports might play out through its effects on reporting biases (especially since its goal was to

**Table 2. Treatment effects of nudges, by reporting source and household remoteness.**

| | Children in employment (self-report) | Children in employment (parent's report) | | |
|---|---|---|---|---|
| | (1) | (2) | (3) | (4) |
| **Panel A - Work in cocoa fields** | | | | |
| Nudge to parents | 0.054 | 0.089* | 0.120** | 0.145** |
| | (0.040) | (0.045) | (0.056) | (0.061) |
| Distance to school (km) | | | | 0.016 |
| | | | | (0.012) |
| Nudge to parents × Distance | | | | -0.061* |
| | | | | (0.031) |
| | | | | |
| Control mean | 0.498 | 0.221 | 0.230 | 0.230 |
| R-squared | 0.115 | 0.091 | 0.094 | 0.096 |
| Observations | 2,246 | 2,176 | 1,608 | 1,608 |
| **Panel B - Domestic work** | | | | |
| Nudge to parents | 0.046** | 0.051** | 0.066** | 0.072** |
| | (0.020) | (0.022) | (0.026) | (0.028) |
| Distance from school (km) | | | | -0.003 |
| | | | | (0.012) |
| Nudge to parents × Distance | | | | -0.015 |
| | | | | (0.031) |
| Control mean | 0.878 | 0.789 | 0.794 | 0.794 |
| R-squared | 0.034 | 0.044 | 0.050 | 0.050 |
| Observations | 2,246 | 2,175 | 1,607 | 1,607 |
| Controls (baseline survey) | Yes | Yes | Yes | Yes |

*Notes*: In column (1), Children in employment (self-report) = 1 if the child answered affirmatively to the working question: "In the last month, have you engaged in one or more of the following activities, for one hour or more? Work in a cocoa plantation" in Panel A, "Domestic work, such as buying food or cooking, cleaning the house, do the laundry, take care of children or other sick or old relatives", in Panel B. In columns (2) to (4), Children in employment (parent's report) = 1 if the parent answered affirmatively to the question "In the last month, has any of your children engaged in one or more of the following activities, for one hour or more? Work in a cocoa plantation", in Panel A, and "Domestic work, such as buying food or cooking, cleaning the house, do the laundry, take care of children or other sick or old relatives", in Panel B. Distance from school (km) is the linear distance between the home and their child's school GPS locations, in kilometers; Nudge to parents = 1 if the parent (but not teachers) was assigned to nudges, and 0 otherwise. Baseline controls include treatments arms, child gender, grade indicators, standardized test scores (averaged across numeracy and literacy); and summary measures of parental engagement, student effort, socio-emotional skills, working memory, visual attention, impulsivity, self-esteem, and mindset (see [14]. Summary measures computed following [17], standardizing each component by normalizing values by the mean and standard deviation of the control group at the baseline survey within each grade, and then averaging over all standardized components. In columns (3) and (4), sample restricted to observations with valid home GPS coordinates. All regressions estimated through Ordinary Least Squares with standard errors clustered at the classroom level. *** p<0.01, ** p<0.05, * p<0.10

openly address the issue without creating stigma). All columns control for baseline characteristics. Because the over-arching intervention evaluated in [14] was centered around teachers, we cluster standard errors at the classroom level.

In Panel A of Table 2, column (1) documents that the intervention did not systematically affect children in employment according to self-reports. In contrast, based on parents' reports, one would have concluded that the intervention *increased* the prevalence of children in employment (column 2). The effect is sizeable: an 8.9 p.p. increase, significant at the 10% level—40.3% of its end-line prevalence in the control group. Column (3) documents that the same holds within the sub-sample with valid home GPS information. Most importantly, column (4) documents that this is likely an artifact of the interplay of the intervention with social desirability biases in communities that have been previously sensitized: very close to the school, the estimated effect size is even larger (a 14.5 p.p. increase, significant at the 5% level),

but it sharply decreases with distance to school, fading out at an approximately 3 km radius, based on our linear estimate.

S3 Table in SI Text further documents that results are robust to outlier observations: the coefficient of nudges to parents in column (4) is virtually unchanged if we winsorize distance from school at the 99th percentile.

Reassuringly, when it comes to domestic work (Panel B), not only were effect sizes very similar regardless of reporting sources (an increase of 4.6 p.p. according to children, in column 1, and of 5.1 p.p. according to parents, in column 2, both significant at the 5% level), but also, treatment effects did *not* vary systematically with distance to the school.

All in all, results are consistent with the interpretation that the social stigma around child labor in cocoa fields (in contrast to domestic work) might introduce bias in official statistics when it comes to evaluating the impacts of policies to discourage child labor whose impacts might at least partially affect its social acceptability—and, hence, magnify or attenuate reporting biases in parents' reports.

## 4 Conclusions

Data on the prevalence of child labor across space and over time is critical for governments and international organizations committed to ensuring children's rights. In the cocoa industry, particularly intensive in child labor given its low rate of mechanization, data issues have posed important challenges to monitoring and enforcement efforts by international organizations and policymakers over the years.

Our finding that child labor statistics following the ILO methodology not only misrepresent the prevalence of child labor, but also mischaracterize its trends (where interventions were in place), raises critical concerns. While the general sentiment of the literature on child labor is that substantial progress has been achieved in recent decades ("[a]n important lesson from all the literature reviewed herein is that child labor can change dramatically and quickly in countries as a result of changes in the economic and policy environment."; [18], pp. 27–28), our results call that sentiment into question. Concretely, in SI Text, we use our estimates to calibrate predictions of the bias-adjusted prevalence of child labor in Côte d'Ivoire and Ghana. Under the stated assumptions, child labor could affect nearly 6.9 million children in these countries, 50% more than in the latest World Development Indicators' statistics. Naturally, this does not invalidate the claim that child labor might be caused by lack of economic development or that it can respond to policy changes [18]; however, it does raise caution about carefully interpreting changes in adult reports in contexts of fast transitions—in light of how such changes might impact not only the economic fundamentals that generate trade-offs between employing children vis-a-vis investing in their human capital, but also, reporting biases themselves (e.g., by changing parents' beliefs about what is socially appropriate).

Based on our results, asking children independently about whether they work (and, if so, how many hours) could yield child labor indicators consistent with costly-verification data. There are, however, technical challenges involved in interviewing children, including whether children are asked inside or outside the household, and variations in the understanding of what exactly characterizes 'work' by children in different countries (see [19] and [6] for a broader discussion of different framing issues). While the experiences of NORC and Tulane University in Côte d'Ivoire and Ghana since 2008–2009 could inform the replication of self-reports elicitation of child labor by other agencies moving forward, surveying children directly involves some ethical challenges as well. In particular, children might be exposed to

violence if interviews trigger backlash from parents, even when they are conducted in a different setting (e.g., schools). These concerns also limit the potential of technologies worn by children, such as GPS trackers, despite their validated accuracy [7].

For those reasons, finding ways to limit reporting biases in adult surveys may be a superior (and necessary) alternative. Progress in this space has, however, been slow. Most studies focus on indirect elicitation methods (i.e., list experiments, in various forms; e.g., [3]), but these methods generate estimates that are typically imprecise and not necessarily closer to the 'ground truth' (see, for instance, https://statmodeling.stat.columbia.edu/2014/04/23/thinking-list-experiment-heres-list-reasons-think/). In contrast, leveraging technologies to survey adults in communities less subject to previous sensitization and during periods when it is easier to observe child labor might be a much better way forward.

Naturally, that also involves its own challenges. Although we have documented that remoteness is not systematically associated with the true prevalence of child labor (based on children's self-reports) in the close vicinity of schools, it might be that the communities surveyed based on satellite imagery are *not* representative of the territories or the populations of interest, leading to biased aggregate estimates of children in employment and child labor. Moreover, if remote sensing through satellite imagery is imperfect (e.g., if the geographical coverage of the available data is selective either when it comes to which regions it covers or which farms it surveys within each region), then the estimates will again be biased. There are also cost considerations. Obtaining access to high-frequency data that can inform prediction models (and hiring staff or external vendors to train and update such models) might be expensive. Such requirements might limit the ability to learn timely about how child labor evolves across space, particularly in response to policy interventions.

While we expect our contributions to generalize beyond the Ivorian cocoa industry, a limitation is that our sample consists entirely of school children – all of whom were enrolled at the time of our surveys. This excludes children who had dropped out of school (or never enrolled in the first place), presumably more likely to work in cocoa fields. While their parents might be less sensitive to social desirability biases (since their children are not in school), there are other potential sources of social pressure to under-report children in employment—particularly economic pressures linked to restrictions to child labor in global supply chains. Investigating if parents' under-reporting changes with children's school enrollment remains a promising avenue for future work.

## Supporting information

**SI Text. Contains S1–S18 Figs and S1–S5 Tables.**
(PDF)

## Acknowledgments

We thank Maxime Deroubaix and Michael Kra from ENVERITAS for explaining the survey methodology in detail and kindly granting access to their geo-referenced child labor data. We also acknowledge helpful comments from Andrew Dillon and Megan Passey (International Cocoa Initiative), excellent research assistance by Laura Ogando, Gabriel de Campos and Thiago da Costa, and field supervision by Nicolo Tomaselli and Innovations for Poverty Action. All remaining errors are ours.

## Author contributions

**Conceptualization:** Guilherme Lichand, Sharon Wolf.

**Data curation:** Guilherme Lichand.

**Formal analysis:** Guilherme Lichand.

**Funding acquisition:** Sharon Wolf.

**Investigation:** Guilherme Lichand.

**Methodology:** Guilherme Lichand.

**Project administration:** Guilherme Lichand.

**Supervision:** Sharon Wolf.

**Visualization:** Guilherme Lichand.

**Writing – original draft:** Guilherme Lichand, Sharon Wolf.

**Writing – review & editing:** Guilherme Lichand, Sharon Wolf.

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
