## [Decision Letter · Decision Letter 0]

12 Jan 2025

PONE-D-24-42933Measuring Child Labor: the Who’s, the Where’s, the When’s, and the Why’sPLOS ONE

Dear Dr. Lichand, 

Thank you for submitting your manuscript to PLOS ONE. After careful consideration, we feel that it has merit but does not fully meet PLOS ONE’s publication criteria as it currently stands. Therefore, we invite you to submit a revised version of the manuscript that addresses the points raised during the review process. Please see all the review comments below this email.

Please submit your revised manuscript by Feb 26 2025 11:59PM. If you will need more time than this to complete your revisions, please reply to this message or contact the journal office at plosone@plos.org. Please include the following items when submitting your revised manuscript:

We look forward to receiving your revised manuscript.

Kind regards,

Osmond Ekwebelem

Academic Editor

PLOS ONE

Journal Requirements:

“Study conducted in partnership with Transforming Education in Cocoa Communities (TRECC) and generously funded by the Jacobs Foundation Science of Learning initiative. Any views and opinions contained in this paper are those of the authors and do not necessarily reflect the views or opinions of TRECC or the Jacobs Foundation”

Please state what role the funders took in the study.  If the funders had no role, please state: "The funders had no role in study design, data collection and analysis, decision to publish, or preparation of the manuscript.” If this statement is not correct you must amend it as needed. 

3. Please note that funding information should not appear in the Acknowledgments section or other areas of your manuscript. We will only publish funding information present in the Funding Statement section of the online submission form. Please remove any funding-related text from the manuscript. 

“Lichand is a partner and chairman at Movva, the implementing partner of the intervention evaluated as part of this study. Wolf has no competing interests to declare.”

Please confirm that this does not alter your adherence to all PLOS ONE policies on sharing data and materials, by including the following statement: "This does not alter our adherence to PLOS ONE policies on sharing data and materials.” (as detailed online in our guide for authors http://journals.plos.org/plosone/s/competing-interests).  

If there are restrictions on sharing of data and/or materials, please state these. Please note that we cannot proceed with consideration of your article until this information has been declared. 

7. We note that Supporting Information Figures A3 and A4 in your submission contain map/satellite images which may be copyrighted. All PLOS content is published under the Creative Commons Attribution License (CC BY 4.0), which means that the manuscript, images, and Supporting Information files will be freely available online, and any third party is permitted to access, download, copy, distribute, and use these materials in any way, even commercially, with proper attribution. For these reasons, we cannot publish previously copyrighted maps or satellite images created using proprietary data, such as Google software (Google Maps, Street View, and Earth). For more information, see our copyright guidelines: http://journals.plos.org/plosone/s/licenses-and-copyright.

1) You may seek permission from the original copyright holder of Figures A3 and A4 to publish the content specifically under the CC BY 4.0 license.  

2) If you are unable to obtain permission from the original copyright holder to publish these figures under the CC BY 4.0 license or if the copyright holder’s requirements are incompatible with the CC BY 4.0 license, please either i) remove the figure or ii) supply a replacement figure that complies with the CC BY 4.0 license. Please check copyright information on all replacement figures and update the figure caption with source information. If applicable, please specify in the figure caption text when a figure is similar but not identical to the original image and is therefore for illustrative purposes only.

Reviewers' comments:

Reviewer's Responses to Questions

**Comments to the Author**

1. Is the manuscript technically sound, and do the data support the conclusions?

Reviewer #1: Yes

Reviewer #2: Partly

Reviewer #3: Yes

2. Has the statistical analysis been performed appropriately and rigorously? 

Reviewer #1: Yes

Reviewer #2: Yes

Reviewer #3: Yes

3. Have the authors made all data underlying the findings in their manuscript fully available?

Reviewer #1: Yes

Reviewer #2: Yes

Reviewer #3: No

4. Is the manuscript presented in an intelligible fashion and written in standard English?

Reviewer #1: Yes

Reviewer #2: Yes

Reviewer #3: Yes

5. Review Comments to the Author

Reviewer #1: Hi Dear Author,

An important subject has been touched upon and it is a beautiful work in terms of design. The suggestion should be refocused and stated more clearly. Explain the practical implication and further research suggestions based on the finding.

Reviewer #2: The topic is good, but need some follow up:

1. In the introductio, it is necessary to dig deeper into previous studies on child labor in terms of the development of measurement methods, child labor trends, the role of parents in increasing or reducing child labor and the success or failure of the government's role in supervising child labor, as well as the influence of remote and non-remote areas on child labor trend. These things can be demonstrated by providing examples of comparisons with child labor levels in other countries in various types of industry that have the potential to use child labor

2. For discussion, it is necessary to add an explanation of the results of the correlation between child labor and test scores and drop outs that shown in the supplementary data. Then, also need to explain the bias in the reporting of parents, children and Enveritas. All three show different results, so it is necessary to explain which one is more accurate. If the researcher suggest that Enveritas can give the better data that other instrument, so please explain and give the evidence that Enveritas can lower the Bias result of Child labor report. Apart from that, there is also a need for discussion supported by literature regarding the finding that the more remote the area, the less child labor. Also, why child labor is more often found for domestic work than plantation work. Then conceptualize the findings in Cote D'Ivore in general regarding the development of child labor globally.

3. For the result, it is necessary to add a GPS image when taking the data which shows that the child labor was indeed photographed with this tool which may be shown in a certain color. So, not only the coverage of cocoa area.

Reviewer #3: Referee Report for PONE-D-24-42933

This paper studies a very important and pressing question: to what extent do parents and children under-report the incidence of child labor? The study focuses on cocoa-producing communities in Cote d’Ivoire ---- a context where child labor is frequently engaged, partially due to the nature of the cocoa production process (where a small child may still have decent productivity).

This study uses multiple sources of data: parents surveys, children surveys, and novel satellite imagery-based data to study the extent of under-reporting of child labor practices. The study offers convincing evidence that parents systematically under-report the incidence of child labor by a wide margin, whereas children provide somewhat accurate report of child labor.

I have a few comments to improve the paper.

1. The analysis described in section 5.2 and Table 1 seems to use only endline data from the survey. I would have preferred to see this analysis done using the baseline data, as the relationship between child-parent reporting discrepancy and distance to school may have been affected by the anti-child labor intervention carried out between baseline and endline? At the very minimum, the authors should provide Table 1 using both baseline and endline data.

2. The analysis described in section 5.2 and Table 1, all regressions assume a linear relationship between “distance to school” and “child labor” / “discrepancy. This assumption seems quite unnatural – especially given that Appendix Figure C3 (distribution of distance to school) shows a very long right tale. The majority of students live within 1km of their school. However, a few live very far away: 5km, 15km, or even 20 km.

a. First, the very high distance numbers are hard to believe, as these are primary school children. If the household is indeed 15 or 20 km away, how does the child commute to school daily? Should these observations at the tail end of the distribution be excluded from the regressions in Table 1? I would like to see a robustness Table excluding the very high distance tables.

b. The linear assumption may also be a stretch. Can the authors present alternative specifications using: (1) log distance to school, and (2) dummy variables showing: 0-0.5km, 0.5-1km, 1-2km, beyond 2km?

3. Appendix Figure D2 shows that teachers are just as likely to under-report child labor as parents. An intriguing finding. The main text completely steers away from any discussion of the teachers’ reporting. Why?

4. It’s great to see Section 2.1 with very clear definitions of child labor and how they are supposed to be measured in various frequently cited data. This is great. However, I found the various statistics cited on Page 1 (first three paragraphs of the Introduction) very confusing.

a. “2020 data from the International Labour Organization (ILO) documented that 160 million children worldwide were active workers – 9.6% of all 5-17 year-old children” –-- by definition, among these 160 million, not all count as child labor? Because not all “active workers” between age 5-17 count as child labor. This sentence made it seems like there were 160 million child labor in 2020 according to ILO estimates. Please clarify.

b. Similarly, for the next two sentences, “in Sub-Saharan Africa, affecting 23.9% of children” –-- does this mean 23.9% of those under 18 engage in child labor (which is very bad)? Or does it mean that 23.9% of those under 18 engage in active work (which may not be bad)?

c. The next sentence: “according to Save the Children, child labor plummeted globally between 2000 and 2016, from 246 to 152 million infant workers.” What is “infant workers”? This term is even more confusing. Please also be clear whenever “child” is mentioned in the Introduction, what age range are we referring to, under 18 or under 15?

d. In the 3rd paragraph: “According to NORC… 38% of 5-17 year olds in Côte d’Ivoire reported to have worked in 2018-19 (Sadhu et al., 2020); in contrast, the ILO figure for 2016 – based on adult reports for children employment – was only 23%” –-- It seems to me that these discrepancies would be expected if NORCs’ 38% refers to percent of 5-17 year olds working (some of which do not account as child labor, e.g. those aged 15-17 working in non-hazardous tasks), whereas the ILO’s 23% is restricted to child labor.

Generally, the Introduction made it seem like “child labor” means any person under 18 working in any capacity. Please clarify the various statistics cited, which ones are child labor, which ones merely describe under-18 workers.

6. PLOS authors have the option to publish the peer review history of their article (what does this mean?). If published, this will include your full peer review and any attached files.

Reviewer #1: No

Reviewer #2: No

Reviewer #3: No

---

## [Author Response · Author response to Decision Letter 1]

10 Mar 2025

[See Rebutall_letter.pdf]

Dear Dr. Ekwebelem,

Thank you very much for the thoughtful feedback on our original submission, and for the opportunity to revise our work for PLOS ONE. We have now carefully revised the manuscript to follow all style requirements of the journal, and to incorporate all comments by the referees.

In particular, we have conducted additional empirical exercises to ensure that our findings are robust to (1) removing outlier observations, and (2) parametric assumptions of our econometric specification.

We have also carefully rewritten large segments of the paper, unifying references to different child labor statistics to avoid confusion, and more carefully characterizing previous findings from the literature in the Conclusions.

All in all, we believe that we have successfully addressed all concerns raised by the referees and by the Editor and that the manuscript is much stronger as a result.

Thank you once again for the attention dedicated to our submission, and for your numerous inputs – which have helped us significantly improve the paper.

We include a point-by-point response to the referees comments below (shown in blue in Rebutall_letter.pdf).

Reviewer #1: Hi Dear Author,

An important subject has been touched upon and it is a beautiful work in terms of design. The suggestion should be refocused and stated more clearly. Explain the practical implication and further research suggestions based on the finding.

Thank you for the encouraging feedback and for the suggestion to make the practical implications of our findings and the directions for future research more clear.

We have compiled below that changes we have made accordingly in the Conclusions section, as follows:

“Data on the prevalence of child labor across space and over time is critical for governments and international organizations committed to ensuring children’s rights. In the cocoa industry, particularly intensive in child labor given its low rate of mechanization, data issues have posed important challenges to monitoring and enforcement efforts by international organizations and policymakers over the years.

Our finding that child labor statistics following the ILO methodology not only misrepresent the prevalence of child labor, but also mischaracterize its trends (where interventions were in place), raises critical concerns. While the general sentiment of the literature on child labor is that substantial progress has been achieved in recent decades (‘[a]n important lesson from all the literature reviewed herein is that child labor can change dramatically and quickly in countries as a result of changes in the economic and policy environment’; Edmonds and Theoharides, 2021, pp. 27-28), our results call that sentiment into question Concretely, Appendix F in Supplementary Materials uses our estimates to calibrate predictions of the bias-adjusted prevalence of child labor in Côte d’Ivoire and Ghana. Under the stated assumptions, child labor could affect nearly 6.9 million children in these countries, 50% more than in the latest World Development Indicators’ statistics. Naturally, this does not invalidate the claim that child labor might be caused by lack of economic development or that it can respond to policy changes (Edmonds and Theoharides, 2021); however, it does raise caution about carefully interpreting changes in adult reports in contexts of fast transitions – in light of how such changes might impact not only the economic fundamentals that generate trade-offs between employing children vis-a-vis investing in their human capital, but also, reporting biases themselves (e.g., by changing parents’ beliefs about what is socially appropriate).

Based on our results, asking children independently about whether they work (and, if so, how many hours) could yield child labor indicators consistent with costly-verification data. There are, however, technical challenges involved in interviewing children, including whether children are asked inside or outside the household, and variations in the understanding of what exactly characterizes ‘work’ by children in different countries (see

Guarcello et al., 2010 and Dillon, 2010 for a broader discussion of different framing issues). While the experiences of NORC and Tulane University in Côte d’Ivoire and Ghana

since 2008-09 could inform the replication of self-reports elicitation of child labor by other

agencies moving forward, surveying children directly involves some ethical challenges as

well. In particular, children might be exposed to violence if interviews trigger backlash from parents, even when they are conducted in a different setting (e.g., schools). These concerns also limit the potential of technologies worn by children, such as GPS trackers, despite their validated accuracy (Dillon et al., 2017).

For those reasons, finding ways to limit reporting biases in adult surveys may be a superior (and necessary) alternative. Progress in this space has, however, been slow. Most

studies focus on indirect elicitation methods (i.e., list experiments, in various forms), but these methods generate estimates that are typically imprecise and not necessarily closer to the ‘ground truth’. In contrast, leveraging technologies to survey adults in communities less subject to previous sensitization and during periods when it is easier to observe child labor might be a much better way forward. Naturally, that also involves its own challenges. Although we have documented that remoteness is not systematically associated with the true prevalence of child labor (based on children’s self-reports) in the close vicinity of schools, it might be that the communities surveyed based on satellite imagery are not representative of the territories or the populations of interest, leading to biased aggregate estimates of children in employment and child labor. Moreover, if remote sensing through satellite imagery is imperfect (e.g., if the geographical coverage of the available data is selective either when it comes to which regions it covers or which farms it surveys within each region), then the estimates will again be biased. There are also cost considerations. Obtaining access to high-frequency data that can inform prediction models (and hiring staff or external vendors to train and update such models) might be expensive. Such requirements might limit the ability to learn timely about how child labor evolves across space, particularly in response to policy interventions.

While we expect our contributions to generalize beyond the Ivorian cocoa industry, a limitation is that our sample consists entirely of school children – all of whom were enrolled at the time of our surveys. This excludes children who had dropped out of school (or never enrolled in the first place), presumably more likely to work in cocoa fields. While their parents might be less sensitive to social desirability biases (since their children are not in school), there are other potential sources of social pressure to under-report children in employment – particularly economic pressures linked to restrictions to child labor in global supply chains. Investigating if parents’ under-reporting changes with children’s school enrollment remains a promising avenue for future work”.

Thank you once again for your inputs, which have helped us greatly improve the paper.

Reviewer #2: The topic is good, but needs some follow-up:

1. In the introduction, it is necessary to dig deeper into previous studies on child labor in terms of the development of measurement methods, child labor trends, the role of parents in increasing or reducing child labor and the success or failure of the government's role in supervising child labor, as well as the influence of remote and non-remote areas on child labor trend. These things can be demonstrated by providing examples of comparisons with child labor levels in other countries in various types of industry that have the potential to use child labor.

Thank you for the encouraging feedback and for the suggestion to more deeply characterize the previous literature on measuring methodologies, and child labor levels and trends.

We discuss different measurement methodologies in detail, both in the Introduction and in the Conclusions section. In the former:

“Previous studies were not able to document the extent of under-reporting by parents. These papers rely on three types of comparisons. First, comparisons between parents’ reports under different conditions for social desirability bias (e.g., Jouvin, 2021), to document that such biases exist – but unable to pin down their magnitude. Second, comparisons between adults’ and children’s reports (Dillon, 2010; Galdo et al., 2020), to document that discrepancies exist – but unable to pin down what the ‘ground truth’ is. Third, comparisons between an objective measure of children in employment and children’s self-reports (Dillon et al., 2017), to document that the latter are accurate – but unable to pin down under-reporting by parents. Similarly to ours, the latter study documents that children accurately report the number of hours they work, using logs from GPS trackers worn by different household members to verify data from surveys and activity diaries. The study, however, actively refrained from having enumerators ask adults about child labor. In the absence of parents’ reports, it could not document whether parents under-report children in employment and, if so, by how much.”

In the Conclusions section, we complement this characterization as follows:

“Based on our results, asking children independently about whether they work (and, if so, how many hours) could yield child labor indicators consistent with costly-verification data. There are, however, technical challenges involved in interviewing children, including whether children are asked inside or outside the household, and variations in the understanding of what exactly characterizes ‘work’ by children in different countries (see

Guarcello et al., 2010 and Dillon, 2010 for a broader discussion of different framing issues). While the experiences of NORC and Tulane University in Côte d’Ivoire and Ghana

since 2008-09 could inform the replication of self-reports elicitation of child labor by other

agencies moving forward, surveying children directly involves some ethical challenges as

well. In particular, children might be exposed to violence if interviews trigger backlash from parents, even when they are conducted in a different setting (e.g., schools). These concerns also limit the potential of technologies worn by children, such as GPS trackers, despite their validated accuracy (Dillon et al., 2017).

For those reasons, finding ways to limit reporting biases in adult surveys may be a superior (and necessary) alternative. Progress in this space has, however, been slow. Most

studies focus on indirect elicitation methods (i.e., list experiments, in various forms), but these methods generate estimates that are typically imprecise and not necessarily closer to the ‘ground truth’. In contrast, leveraging technologies to survey adults in communities less subject to previous sensitization and during periods when it is easier to observe child labor might be a much better way forward. […]”

When it comes to child labor levels and trends, we agree that the previous version of the manuscript introduced some confusion due to the multiplicity of measures mentioned in the introduction. We have now re-written the Introduction to remove ambiguity, as follows:

“Child labor is a pervasive practice in agriculture, especially in West Africa, where the global cocoa industry sources roughly half of its produce. Defined according to international standards and national legislation, child labor typically characterizes any work by children under the age of 12, and excessive or risky work by 12-17 year-olds (see Section 2.1.1). 2020 data from the International Labor Organization (ILO) documented 160 million child workers worldwide – 9.6% of all 5-17 year-old children (ILO and UNICEF, 2021). Strikingly, that figure was over 2-fold in Sub-Saharan Africa (23.9%). In Côte d’Ivoire, the setting of our study, 15% of the cocoa industry employees that year were children (Sadhu et al., 2020).”

To avoid confusion while still being able to reference commonly cited statistics on child labor levels and trends – albeit measured in different geographies and using different methodologies –, we moved those descriptions to a new subsection within Methods and materials (‘How child labor is measured’), as follows:

“There are striking differences between children’s and adults’ reports in the few settings where both have been asked independently about children in employment. According to NORC (a research institute based at the University of Chicago), which surveys children directly about the number of hours worked in cocoa fields, 38% of 5-17 year olds in Côte d’Ivoire reported to have worked in 2018-19 (Sadhu et al., 2020); in contrast, the ILO figure for 2016 – based on adult reports for children employment – was only 23% (ILO and UNICEF, 2021). Even worse, different sources tell very different stories about child labor’s recent trends. While ILO data indicate a 38% decrease in child labor worldwide since 2000 (Christian Science Monitor, 2019), NORC data record a nearly 65% increase in child labor since 2008-09 (Sadhu et al., 2020).

While these differences are suggestive that official statistics based on adult reports are biased, it is hard to be sure; besides differences in whom is surveyed, discrepancies across surveys might also accrue to differences in their geographical coverage or in the timing of data collection, or to other methodological differences. Moreover, in the absence of verification, it is unclear whether children’s self-reports do not suffer from reporting biases too.”

While it is hard to comment on government’s success or failures in supervising child labor (in remote areas or otherwise) given such data reliability issues, we now added a more explicit take in the Conclusions section, as follows:

“Our finding that child labor statistics following the ILO methodology not only misrepresent the prevalence of child labor, but also mischaracterize its trends (where interventions were in place), raises critical concerns. While the general sentiment of the literature on child labor is that substantial progress has been achieved in recent decades (‘[a]n important lesson from all the literature reviewed herein is that child labor can change dramatically and quickly in countries as a result of changes in the economic and policy environment’; Edmonds and Theoharides, 2021, pp. 27-28), our results call that sentiment into question Concretely, Appendix F in Supplementary Materials uses our estimates to calibrate predictions of the bias-adjusted prevalence of child labor in Côte d’Ivoire and Ghana. Under the stated assumptions, child labor could affect nearly 6.9 million children in these countries, 50% more than in the latest World Development Indicators’ statistics. Naturally, this does not invalidate the claim that child labor might be caused by lack of economic development or that it can respond to policy changes (Edmonds and Theoharides, 2021); however, it does raise caution about carefully interpreting changes in adult reports in contexts of fast transitions – in light of how such changes might impact not only the economic fundamentals that generate trade-offs between employing children vis-a-vis investing in their human capital, but also, reporting biases themselves (e.g., by changing parents’ beliefs about what is socially appropriate).”

2. For discussion, it is necessary to add an explanation of the results of the correlation between child labor and test scores and drop outs that shown in the supplementary data. Then, also need to explain the bias in the reporting of parents, children and Enveritas. All three show different results, so it is necessary to explain which one is more accurate. If the researcher suggest that Enveritas can give the better data that other instrument, so please explain and give the evidence that Enveritas can lower the Bias result of Child labor r

---

## [Editor Report · Decision Letter 1]

1 Apr 2025

Measuring Child Labor: the Who’s, the Where’s, the When’s, and the Why’s

PONE-D-24-42933R1

Dear Dr. Lichand,

We’re pleased to inform you that your manuscript has been judged scientifically suitable for publication and will be formally accepted for publication once it meets all outstanding technical requirements.

Kind regards,

Osmond Ekwebelem

Academic Editor

PLOS ONE
---

## [Editor Report · Acceptance letter]

PONE-D-24-42933R1

PLOS ONE

Dear Dr. Lichand,

I'm pleased to inform you that your manuscript has been deemed suitable for publication in PLOS ONE. Congratulations! Your manuscript is now being handed over to our production team.

Kind regards,

on behalf of

Dr. Osmond Ekwebelem

Academic Editor

PLOS ONE